# Understanding the Palatability, Flavor, Starch Functional Properties and Storability of Indica-Japonica Hybrid Rice

**DOI:** 10.3390/molecules27134009

**Published:** 2022-06-22

**Authors:** Xue Gong, Lin Zhu, Aixia Wang, Huihan Xi, Mengzi Nie, Zhiying Chen, Yue He, Yu Tian, Fengzhong Wang, Litao Tong

**Affiliations:** 1Key Laboratory of Agro-Products Processing Ministry of Agriculture, Institute of Food Science and Technology, Chinese Academy of Agricultural Sciences, Beijing 100193, China; xuegong1214@163.com (X.G.); zhk960656147@163.com (A.W.); 18437906881@163.com (H.X.); niemengzi315@163.com (M.N.); 15989109538@163.com (Z.C.); hy1257585181@163.com (Y.H.); tianyu152302@163.com (Y.T.); 2Key Laboratory of Preservation Engineering of Agricultural Products, Ningbo Academy of Agricultural Sciences, Institute of Agricultural Products Processing, Ningbo 315040, China; zhulin0822@163.com

**Keywords:** starch-based food, indica–japonica hybrid rice, palatability, aroma components, starch functionality, storability

## Abstract

The rice quality and starch functional properties, as well as the storability of three YY-IJHR cultivars, which included YY12 (biased japonica type YY-IJHR), YY1540 (intermedius type YY-IJHR) and YY15 (biased indica type YY-IJHR), were studied and compared to N84 (conventional japonica rice). The study results suggested that the three YY-IJHR varieties all had greater cooking and eating quality than N84, as they had lower amylose and protein content. The starch of YY-IJHR has a higher pasting viscosity and digestibility, and there was a significant difference among the three YY-IJHR cultivars. Rice aroma components were revealed by GC-IMS, which indicated that the content of alcohols vola-tile components of YY-IJHR were generally lower, whereas the content of some aldehydes and esters were higher than N84. In addition, YY-IJHR cultivars’ FFA and MDA contents were lower, which demonstrated that YY-IJHR had a higher palatability and storability than those of N84 in fresh rice and rice stored for 12 months. In conclusion, this study suggested that YY-IJHR had better rice quality and storability than N84. PCA indicated that the grain quality and storability of YY12 and YY15 were similar and performed better than YY1540, while the aroma components and starch functional properties of YY-IJHR cultivars all had significant differences.

## 1. Introduction

Rice (*Oryza sativa*) with abundant starch, is a globally important starch-based food that provides calories to over 60% of the world’s population [1]. To meet the rice demand of large populations worldwide, breeders around the world make use of the strong heterosis between indica and japonica rice to improve the rice yield. In recent years, the breeding of indica–japonica hybrid rice has developed very quickly and some cultivars with high yield have successfully been released. Yongyou indica–japonica hybrid rice (YY-IJHR), a new hybrid rice with high yield, is widely cultivated in southern China [2]. At the city of Ningbo in Zhejiang Province, China, the mean grain yield of Yongyou11, Yongyou12, Yongyou13, Yongyou15, Yongyou538, Yongyou1538, Yongyou1540 and Yongyou4540 were all over 12.0 t/ha, which was significantly higher than conventional rice [3,4]. Most studies on YY-IJHR focus on cultivation techniques and yield [4,5,6,7]. In recent years, there have been preliminary studies on the characteristics of indica–japonica hybrid rice starch [8]. With the continuous expansion of planting area, the rice quality and storage characteristics of YY-IJHR have attracted increasing attention from rice industrialists, physiologists and consumers. 

Rice quality is reflected by the nutritional components and eating quality, as well as the aroma compounds. Recently, the report of Bian et al. [9] suggested that different panicle types of indica–japonica hybrid cultivars had a different yield and grain quality. Starch, as the major component of rice, as well as its functional properties, play an important role in determining rice’s edible quality and end-uses [10]. The study of Wani et al. [11] suggested that the starch determines the eating and cooking properties of rice grains, with the amylose content particularly affecting the taste value of cooked rice. Zhu et al. [8] indicated that the structural and physicochemical properties of indica–japonica hybrid rice starches were significantly different. In addition to making starch-based food, rice starch is also used in various processed foods as an adhesive, thickener, extending agent, and inflating agent, according to its properties [12]. However, the rice starch research is mainly focused on conventional or hybrid indica, japonica or wild rice [13,14]. The knowledge of YY-IJHR’s starch properties is limited [8]; there was no information regarding comparative studies on the starch functionality of YY-IJHR before and after storage. Furthermore, the type and quantity of aroma compounds were also important indicators of rice quality, which made an important contribution to the taste value of rice. The gas chromatography–ion mobility spectrometry (GC-IMS), as a sensitive and efficient approach, had been increasingly applied to investigate the volatiles in rice, such as Guangxi fragrant rice [15]. However, as a new cultivar, little work has been performed on the aroma volatiles of YY-IJHR, especially the study determining the YY-IJHR volatiles by GC-IMS. In this study, we determined the aroma components of YY-IJHR using GC-IMS and analyzed the characteristic aromas of different cultivars by principal component analysis (PCA). To comprehensively describe the rice quality of YY-IJHR, we evaluated and comparatively analyzed three YY-IJHR cultivars’ characteristic aroma volatiles, eating quality and starch functional properties.

As is well known, the storage of rice inevitably leads to changes in cooking quality, pasting properties and biochemical indicators. The study of Faruq, et al. [16] showed that ageing could influence kernel expansion, water absorption and gelatinization temperature, which ultimately affect the internal structure and cooking quality of rice grains. Another study indicated that storing rice for from 3 to 4 months after harvesting would have a great impact on the cooking quality of grains, maturing the rice and making it taste better [17]. However, to our knowledge, there was little information about the changes in the major cooking quality and pasting property parameters of YY-IJHR under 12 months’ storage. Furthermore, a study has shown that the biochemical indicators regarding the storability of rice stored for one year obviously changed, such as an increase in FFA content and a decrease in antioxidant enzymes’ activity [18]. However, rice storability is a complex trait, which is influenced by many aspects, such as seed genotype, endogenous enzymes activity and storage conditions [19]. Knowledge of rice storability also plays a guiding role in rice uses and consumer choice. As a new rice variety, the storability of YY-IJHR cultivars is not clear. 

An understanding of rice quality, the functional properties of rice starch and rice storability is of critical importance to optimize industrial applications and allow for consumers to select suitable rice varieties [12]. To the best of our knowledge, there was no information systematically evaluating the rice properties of YY-IJHR. Therefore, the objectives of this study were: (i) determine the rice quality, including edible quality and aroma components of YY-IJHR; (ii) examine the functional properties of rice starches before and after 12 months’ storage; (iii) explore the storability of YY-IJHR. Such information may not only contribute to the understanding of rice quality, starch functional properties and the storability of YY-IJHR, but also provide necessary knowledge for consumers, the future work of rice breeding and better applications in the rice industry.

## 2. Results and Discussion

### 2.1. Rice Quality

#### 2.1.1. Rice Nutritional and Eating Quality

Amylose determines the eating and cooking properties of rice grains and protein is typically the second-highest component in many rice-starch based foods, at about 4–20% by weight [10]. The breeding of high-eating-quality rice has been focused on obtaining low amylose and protein content [11]. To evaluate the nutritional and eating quality of YY-IJHR, amylose and protein content were determined. As shown in Figure 1A, the amylose content was higher in N84 (21.65%) compared with the YY-IJHR cultivars (16.45–19.2%). The protein content of N84 was 8.85%, while the protein content of YY-IJHR cultivars was in the range 8.05~8.30% (Figure 1B). The nutritional quality of hybrid rice substantially differed from that of conventional japonica rice, because hybrid rice was an F2-segregated population derived from hybrid F1. Theoretically, there was a greater separation in the indica–japonica hybrid rice (IJHR) of the hybrid F2 population. For YY-IJHR varieties, both amylose and protein contents were decreased compared to those of its parents, which might be explained by the character segregation brought by indica–japonica hybrid genes in the YY series.

The edible quality of rice was mainly reflected by its taste value and freshness [12]. As shown in Figure 1C, the taste values of YY12, YY1540, and YY15 were 79.00, 77.76, 81.36, respectively; which were also higher than that of N84 (74.76). Previous studies showed the rice amylose and protein contents were negatively correlated with sensory quality [13]; these data may explain the higher eating quality of YY-IJHR cultivars compared to N84. Freshness was an important palatability characteristic of rice grain. As shown in Figure 1D, the freshness scores of YY12, YY1540, and YY15 were 65.66, 69.00, 64.33, respectively; all of these were higher than that of N84 (63.3). This result indicated that YY-IJHR had a better palatability than N84. 

Taken together, these data suggest that YY-IJHR varieties have an advantage over N84 in terms of edible quality and YY15 had the best taste value, while YY1540 had the best freshness among the three YY-IJHR cultivars.

#### 2.1.2. The Appearance and Textural Characteristic

The grain length of YY-IJHR cultivars was measured and compared with that of N84. As shown in Figure 1E, the average lengths of YY12, YY1540, and YY15 were 5.3 mm, 5.7 mm and 6.3 mm, respectively, all of which were longer than that of N84 (5.0 mm). Generally, the shape of common indica rice was long and thin, whereas the shape of conventional japonica rice was short and circular. YY-IJHR was a cultivar containing an indica–japonica hybrid genotype, so its length was longer than that of conventional japonica (Figure 1I). The head rice rate, an important factor affecting the sensory evaluation of rice, was considered a major determinant of quality grades in some developing countries [20]. As shown in Figure 1F, the head rice rate of N84 was significantly higher than that of the YY-IJHR cultivars. Therefore, improving the head rice rate might be a problem for the next YY series of indica–japonica hybrid rice breeding. 

Texture is a multi-faceted sensory property, with hardness and stickiness being the most common determined and discriminable textural properties of cooked rice [21]. As shown in Figure 1G,H, there were significant differences in terms of hardness and stickiness between N84 and YY-IJHR cultivars. The hardness of N84 (3.95 kgf) was higher than YY-IJHR cultivars in the range 3.24 kgf~3.36 kgf, while the stickiness of YY-IJHR varieties (0.22 kgf~0.23 kgf) was higher compared with N84 (0.19 kgf). There was no significant difference in hardness and stickiness among YY-IJHR cultivars. The stickiness of cooked rice is important for eating quality and consumer acceptance [21]. Rice stickiness had a significant correlation with leached amylopectin amount, which increased with the decrease in amylose content [21]. Since the hardness was also positively correlated with the amylose content, the low hardness and high stickiness could be explained by the lower amylose content of YY-IJHR. Furthermore, the study of Peng et al. [22] indicated that the taste value of the hybrid rice was negatively correlated with hardness, while it was positively correlated with stickiness. These results were consistent with the results regarding edible quality in this study.

#### 2.1.3. PCA of Rice Quality

The rice-quality characteristics among the four rice varieties was summarized in the scores plot (Figure 1J). The PCA of grain-quality parameters and samples with different rice varieties suggested that PC1 and PC2 accounted for 87.8% of the total variation. Therefore, the plane of PC1–PC2 could reflect the main contributions of the response variables. The similarities and differences between the varieties are determined by the locations of the dots, and the distances of the dots from the arrows can explain the relationship between the two in a biplot. In Figure 1J, all the samples could be divided into two groups according to their corresponding grain quality. Group 1 was the control group (N84), which exhibited high amylose content, high protein content and high hardness values. This result indicated that the high amylose and protein contents could lead to the low taste value and hard texture of cooked rice, which was consistent with the previous study [16]. As shown in Figure 1J, YY12, YY1540, and YY15 formed group 2, which possessed high stickiness and freshness as well as good taste values. This result suggested that YY-IJHR varieties had a better grain-quality performance than N84, the conventional japonica rice.

### 2.2. Analysis of Aroma Components of YY-IJHR

#### 2.2.1. Odor Description of Detected Aroma Components

Previous studies have demonstrated that flavor fingerprints by GC-IMS provide useful information and, therefore, could be used to evaluate the characteristic aroma of grain products [23,24]. The identified visual spots were presented in Gallery Plot. As shown in Figure 2A, the row represented sample and the column represented substance. The color and brightness of the signal peak represents the concentration of the substance, which could directly show the complete volatile compounds’ information of each sample and the difference in volatile compounds between samples. Among the 65 detected peaks (P), a total of 49 volatile components and 16 unknown substances were identified, including alcohols (9P), esters (6P), aldehydes (26P), ketones (7P) and heterocyclic substance (1P), among which 18 pairs were monomers and dimers (the chemical formula and CAS number were the same, but the morphology was different). There were 16 aroma substances that could not be characterized, which resulted from the imperfect information in the flavor composition database. These volatile components all occurred in the YY-IJHR and N84, while the different concentrations of volatiles brought variations in aroma. Among the experimentally detected volatile components, alcohols and esters were the characteristic substances in rice aroma components, and 1-octen-3-ol, n-hexanol, n-pentanol, 2-methylbutanol, and isoamyl alcohol were detected in all rice samples. These alcohols were C4–C8 low-carbon chain alcohols with an alcohol aroma, such as 1-octene-3-alcohol with mushroom and hay aroma, n-hexanol with an apple-like fruit aroma, etc. The detected esters included ethyl acetate, butyl acetate and butyl propionate, which were volatile components with a fruit aroma.

Aldehydes and ketones containing carbonyl groups were important components of rice volatiles. Most of the ketones detected in this study were low-carbon ketones, of which C5–C8 ketones had a special aroma. For example, methyl heptanone had a lemongrass aroma and 2-heptanone had a lemongrass aroma. This had a pear fragrance, while ketones of C4 and below had no fragrance. For example, 2-butanone had a similar smell to acetone. Aldehydes were intermediates that convert alcohols into acids. Aldehydes with a low molecular weight (below C7) have an unpleasant pungent odor, such as butyraldehyde, glutaraldehyde and furfural. With the increase in molecular weight, C8–C10 aldehydes have an obvious floral or fruit aroma, such as nonanal, with a rose and citrus aroma; Octanal had a strong orange aroma. The other detected organic compounds were mainly 2-pentylfuran, which were oxygenated heterocyclic compounds with fruit and grass aromas [25].

#### 2.2.2. Differential Aroma Compounds

As shown in Figure 2A, the characteristics of volatiles among the different rice samples with circled volatile compounds (a1, a2, a3, b1, c1, c2, d1, d2, d3) are presented. The dark red spots indicated the obviously higher concentrations of volatile compounds than the reference. It could be seen that the relative content of alcohols in the YY-IJHR varieties were significantly lower compared with N84, such as hexanol, amyl alcohol, 2-methylbutanol, and isoamyl alcohol. However, the levels of some aldehydes and fruity esters, such as benzaldehyde, furfural, 2-methylbutanal, 3-methylbutanal, ethyl acetate, and butyl acetate, were much higher in YY-IJHR than those in the control group, especially benzaldehyde and butyl acetate. 

There were also significant differences in volatile flavor components among different varieties of YY-IJHR. The volatile flavor components of YY15 were mainly aldehydes and esters, whereas the YY12 were mainly aldehyde and ketone. Differential aroma compounds are landmark substances, which distinguish them from other samples [26]. Combined with ROAV (Figure 2B), 3-methylbutan-1-ol was referred to as the differential in N84 and YY12, while nonanal-D were referred to as differential volatiles in YY1540 and YY15 among the volatile compounds.

#### 2.2.3. Key Aroma Compounds

The relative concentration of volatile flavor components was an important factor in distinguishing rice aroma. The contribution of each compound to the fresh rice flavor was evaluated by calculating the relative odor activity value (ROAV). An ROAV compound of greater than 1 indicated that compounds made a significant contribution to the flavor, and a value < 1 meant a small contribution to the flavor. As shown in Figure 2B, the top 10 volatile compounds of the ROAV value (ROAV > 1) of fresh rice grains were referred to as the main volatile compounds. This meant that the detected volatile compounds made a significant contribution to the aroma. Hexanal-M,2-methylbutanal-D,3-methylbutanal-D, nonanal-M,3-methylbutanal-M,2-methylbutanal-M,1-Octen-3-ol, ethyl-hexanoate, (E)-2-hexenal-M were the common main volatile compounds of the four samples.

The vital compounds that formed the flavor of fresh rice comprised differential volatile compounds and main volatile compounds. Therefore, the key aroma compounds were defined by these two major components [27]. Hexanal-M, 2-methylbutanal-D, 3-methylbutanal-D, nonanal-M, 3-methylbutanal-M, 2-methylbutanal-M, 1-Octen-3-ol, ethyl-hexanoate, (E)-2-hexenal-M, 3-methylbutan-1-ol and nonanal-D were regarded as key aroma compounds in YY-IJHR. 

#### 2.2.4. PCA of Aroma Compounds

A PCA performed on the rice aroma compounds and samples of different rice varieties showed that PC1 and PC2 accounted for 78.6% of the total variation. As shown in Figure 2C, the distance between YY-IJHR group and the control group(N84) was relatively high, indicating that the difference in aroma properties between the YY-IJHR group and control group (N84) was significant. The control group was positive with PC1, whereas the YY-IJHR group was negative. This result could be attributed to the relative, higher concentration of aldehydes, and the ROAV of alcohols was lower in all YY-IJHR samples (Figure 2B). 

The volatile flavor components of YY15 were mainly aldehydes and esters, while those of YY12 were mainly aldehydes and ketones (Figure 2A). As shown in Figure 2A, these aroma compounds’ concentrations were relatively low. Notably, the aroma compound concentration of YY1540 was relatively high. Therefore, the cultivars of YY12 and YY15 were negative with PC2, while YY1540 was positive (Figure 2C). These results suggested that there was a significant difference among three YY-IJHR cultivars and N84, which was revealed by the PCA plot.

### 2.3. Starch Functional Properties of YY-IJHR

Rice starch is used in various processed foods as an adhesive, thickener, extending agent, and inflating agent. The functional properties of rice starch are influenced by the crystalline structure, the amylose–amylopectin ratio and the fine structure of amylopectin [12]. This is important for the rice-starch-based food quality and other end-use utilizations. The digestive properties of cooked rice are affected by the composition of starch, such as rapidly digestible starch and slowly digestible starch, whereas the eating and cooking quality of rice is influenced by the pasting properties of starch [28].

#### 2.3.1. Starch Digestive Property

The digestive properties of rice starch have important potential implications for human health. Starch can be grouped into rapidly digestible starch (RDS), slowly digestible starch (SDS) and resistant starch (RS) [29]. Rapidly digestible starch (RDS) rapidly hydrolyses into sugar molecules after consumption and provides the energy needed by the body. Slowly digestible starch (SDS) is that fraction of starch that is slowly hydrolyzed to glucose molecules in small intestine for the maintenance of normal blood sugar levels [29]. Resistant starches (RS) are those that, by localization, physical, or chemical causes, are unavailable for enzymatic attack, acting as dietary fiber in our organism [30]. 

RS was positively correlated with amylose content in rice starch, while SDS and RDS decreased as amylose content increased [31]. The study of Noda et al. [32] indicated that the amylose content in rice starch was negatively correlated with digestibility because amylose in starch granules was not easily digested by enzymes. As shown in Figure 1A, the amylose content was significantly higher in N84 (21.65%) compared with the YY-IJHR cultivars (16.45–19.2%). This result indicated that the cultivar of N84 had more RS and lower digestibility compared to YY-IJHR.

#### 2.3.2. The Pasting Properties

The characteristic pasting parameter values of rice have a great impact on the properties (e.g., cooking and eating quality, texture) of many rice-starch-based food matrices [10]. The pasting parameters, including peak viscosity (PV), trough viscosity (TV), breakdown (BD), final viscosity (FV), setback (SB) and pasting temperature (PT), are displayed in Table 1. PV reflects the water-holding capacity or swelling degree of the starch granules and is often related to the final product quality, because swollen and collapsed starch granules can affect the texture of the product [11]. The retrograded viscosity (BD) characterizes the thermal and shear resistance of the starch pastes, while the SB shows the retrogradation trend [13]. High BD values indicate high cooking quality because rice degrades to a small extent after cooling [13]. As shown in Table 1, PV, FV, BD of YY-IJHR were all significantly higher than those of N84. The higher PV of YY-IJHR represented the higher swelling capacity because of its lower amylose content, which is consistent with the study of Kong et.al. [13]. The underlying origin of the differences in viscosity was the composition of starch and the fine starch structure in the rice grain. The relatively higher FV of the three YY-IJHR cultivars could be explained by the fact that, with decreasing amylose content and the proportion of short branches of amylopectin, leading to a higher FV [21]. Moreover, studies have indicated that the low amylose content and high PV and BD were characteristics of a high eating quality [33]. Combining the results of the previous research [8,33], it can be concluded that YY-IJHR possessed a higher pasting viscosity and better eating quality than N84. In addition, YY15 had a relatively better eating quality among the three YY-IJHR varieties, in accordance with the results of a lower amylose content and higher taste value.

During storage, the changes in pasting property parameters occurred due to the variation in granule size distribution, amylose content, and crystallinity. As shown in Figure 3A, a greater change in pasting properties (PV, BD, FV, SB) was exhibited in N84 than YY-IJHR after 12 months of storage, which indicated that YY-IJHR has a stronger starch function retention than N84. 

Interestingly, a decreasing trend of PV was shown in YY1540, whereas an increasing trend was shown in the other three varieties during storage. PV represents the water-binding capacity of starch granules [8] and the higher amylose content in rice starch decreased PV [34]. Therefore, the decreasing trend of PV in YY1540 may be due to the increase in amylose during storage, which inhibited the swelling of starch granules, resulting in the decrease in PV. This phenomenon suggested that the stored rice of YY1540 was of relatively lower quality in rich-starch food or swelling agents [12].

#### 2.3.3. PCA of Pasting Properties

To comprehensively analyze the pasting properties, data including PV, TV, BD, FV, SB, PT of fresh rice and rice stored for 12 months among the four cultivars were loaded on PCA and are summarized in the scores plot (Figure 3B). The points in the confidence ellipses were all statistically significant. The PCA of pasting property parameters (Table 2) and different rice cultivars indicated that PC1 and PC2 accounted for 86.3% of the total variation. The samples could be divided into two groups according to their contribution to the principal components. Group 1 was the conventional japonica rice (N84), which was significantly negative with the PC2, which provided the lowest pasting viscosity, in accordance with Figure 2A. Group 2 was consisted of YY12, YY1540 and YY15. As shown in Figure 2A, the PV of YY15 was the highest in both fresh rice and stored for 12 months, while the mean rice starch of YY15 has the best swelling capacity among the four cultivars. The PV of stored rice in YY1540 was significantly decreased, which meant that the starch of YY1540 had a lower retention capacity regarding starch’s functional properties. As shown in Figure 2B, YY15 was significantly positive with PC2 and made the greatest contribution to PC1, while YY1540 was negatively correlated with PC1. Three YY-IJHR cultivars showed significant variation in pasting properties, which was in accordance with the previous report [8]. YY15 possessed a higher pasting viscosity, followed by YY12 and YY1540; N84 had the lowest pasting viscosity.

### 2.4. The Cooking Quality

#### 2.4.1. The Cooking Quality of Fresh Rice

The cooking quality of rice was an important indicator of consumer acceptance. To evaluate the cooking quality of fresh rice in YY-IJHR, the cooking expansion rate, cooking elongation rate, soaked water absorption rate and heating water absorption rate were measured and compared with N84. As shown in Table 2, the cooking expansion rate (322–364%), heating water absorption rate (143.40–151.75%) and soaked water absorption rate (23.87–38.60%) of fresh rice in YY-IJHR cultivars were greater than those of N84 (cooking expansion rate (268%), heating water absorption rate (141.62%), soaked water absorption rate (16.08%). The hydration of rice was influenced by amylose content in rice grain. Varieties with a low amylose content tended to be sticky and cohesive when cooked, absorbed more water, and thus had a greater volume after cooking [35]. Furthermore, the study of Zhu et al. [36] reported that protein could compete with starch to absorb water and further restrict the swelling of starch granules, which may affect the rice expansion and extension rate when heated or soaked with water. Therefore, YY-IJHR cultivars with low amylose and protein content had a greater cooking expansion rate and heating water absorption rate. The greatest heating and soaked water absorption rate (151.75% and 30.86%, respectively) occurred in YY15 among the three YY-IJHR cultivars, which was consistent with its lower amylose content. 

The swelling ability of starch granules when heated with water to cook rice could be reflected by the PV [9]. The YY1540 cultivar has the greatest cooking expansion rate (364%) and elongation rate (80.16%) among the three YY-IJHR cultivars, in accordance with the relatively higher PV in YY1540. To improve consumer satisfaction, the information provided by this study could be used by the rice breeders and industrialists to use these rice varieties’ characteristics as evidence to breed new rice with a greater cooking quality and better rice starch-based foods.

#### 2.4.2. Comparison of the Changes in Cooking Quality during Storage

Ageing also enhances the volume expansion and water absorption of rice upon cooking, resulting in a product with a firmer texture and less stickiness [6]. In this study, the magnitude of changes was also different in the rice cultivars after 12 months of storage. It could be found that the range of changes in cooking quality (cooking expansion rate (3–45%) and heating water absorption rate (1–3%)) of YY-IJHR was smaller than that of N84 (cooking expansion rate (198%), heating water absorption rate (203%)) during storage. The steric hindrance effect of protein and starch–protein interactions during storage might also influence the cooking quality. However, the surrounding structure of protein or the bond between starch and protein were destroyed by the protease, allowing for more water to seep into the starch granules. The higher the protease content, the greater the expansion and extension of rice molecules [36]. Therefore, this greater change magnitude in the cooking quality of N84 could be attributed to the relatively higher protein and amylose content, as well as the high protease activity in N84. Moreover, the range of changes in the four cooking indexes in YY1540 was the smallest after storage. In conclusion, the results of this study suggested that YY-IJHR cultivars had a better cooking-quality retention capacity than N84, especially the YY1540 cultivar.

### 2.5. The Rice Storability

#### 2.5.1. The Content of FFA and MDA

The free fatty acid (FFA) content, a sensitivity indicator of the quality changes, is often employed as a measure of the deterioration of stored grain. As shown in Figure 4A, for the fresh rice, N84 (12.00 mg/100 g) had a similar amount of FFA to YY12 (12.89 mg/100 g), and both were significantly higher than YY1540 (8.80 mg/100 g) and YY15 (8.70 mg/100 g). In addition, the FFA content of N84 (15.98 mg/100 g) significantly increased, becoming higher than that of YY-IJHR after one year of storage. The increase in FFA content in stored grain could be attributed to the hydrolysis of lipase during storage. This can impact the physical properties of rice in terms of its eating quality, flavor, compositions. With a prolongation of storage time, FFA content increased and the quality of stored grain decreased simultaneously [37]. The FFA content was an effective indicator of a deterioration in palatability in the old rice [38]. The lower FFA content of YY-IJHR meant that the quality and palatability of YY-IJHR were better than these of N84 after 12-months storage. 

Malondialdehyde (MDA) is an end-product of seed lipid peroxidation and its content may represent the degree of rice seed cell damage [39]. To explore the storability of YY-IJHR, we determined the MDA content of rice grain in YY-IJHR. As shown in Figure 4B, for fresh rice, the MDA content of YY1540 (21.36 nmol/g) was the highest among the four varieties. The MDA content of YY12 showed a decreasing trend, while the other two YY-IJHR varieties showed an increasing trend after 12 months of storage. This phenomenon occurred because cells began to die, enzyme activity decreased with the extension of storage time, and FFA oxidation significantly reduced, resulting in a decline in the MDA content [39]. Moreover, the MDA content of N84 substantially increased to 46.31 nmol/g, which was significantly higher than that of YY12 (9.80 nmol/g), YY15 (19.00 nmol/g) and YY1540 (29.00 nmol/g) after 12-months’ storage (Figure 4B). The increase in MDA level indicated that the degree of membrane lipid peroxidation in rice increased and rice quality decreased. This result suggested that the storability of YY-IJHR cultivars was higher than N84, which was in accordance with the result of the lower FFA content in YY-IJHR.

In conclusion, both YY-IJHR cultivars’ FFA and MDA contents were lower, which indicated that YY-IJHR had a lower deterioration in palatability and higher storability than N84 over the same storage time.

#### 2.5.2. The Activity of CAT and POD

Catalase (CAT) and peroxide (POD) are rice-seed-viability-protecting enzymes that scavenge hydrogen peroxide in rice seeds. To explore the rice-seed-protecting enzymes’ activity, we determined the CAT and POD activity of fresh rice and rice stored for a year. As shown in Figure 4C,D, for fresh rice, the CAT activity of YY-IJHR in the range 4.15 U/g~6.80 U/g was higher than N84 (3.28 U/g). This was consistent with the results of lower FFA and MDA content in YY-IJHR in this study. After 12 months of natural storage, the CAT and POD activities of all four rice varieties showed a decreasing trend. The decline in the activities of CAT and POD, resulting in a reduced ability to break down the peroxides that are hazardous to cells in the rice grain, thus accelerating the deterioration in rice quality. The decreasing degree of enzyme activity of the four rice varieties was similar, so we judged the storability by the PCA.

#### 2.5.3. PCA of Storability

The relationship between rice cultivars with different aging indexes and antioxidant enzymes was summarized in the biplot (Figure 4E). The PCA of storability indexes and different rice varieties suggested that PC1 and PC2 accounted for 87.8% of the total variation. The similarities and differences between varieties were determined by the point locations and the distances of the points from the arrows may explain the relationship between the two in the biplot. As shown in Figure 4E, the storability of N84 and YY-IJHR were clearly distinguished, as the FFA content of N84 was high before and after storage, and its higher MDA content after storage indicated the inferior storability of N84. In addition, YY12 and YY15 showed similar storage characteristics and had a higher CAT activity in grain compared to N84, which led to the higher free-radical scavenging capacity and lower palatability deterioration.

## 3. Materials and Methods

### 3.1. Materials

Rice seeds were obtained from Shipu Town, Xiangshan County, Ningbo City, Zhejiang Province, China. N84 seeds were used as the high-quality rice control. YY15 (biased indica-type indica–japonica hybrid rice,) and YY1540 (intermedius-type indica–japonica hybrid rice) were harvested from the same field at the end of October. YY12 (biased japonica-type indica–japonica hybrid rice) was harvested around 15 days later due, to its late-maturing character. Fresh rice in the current season were dried to a moisture content of 14–15% after threshing. The rice was packaged into a universal transparent preparation bag and stored in the dark at 4 °C before the formal start of the experiment. The samples were stored at room temperature (26 °C) at the formal start of the experiment. 

### 3.2. Preparation of Rice Flour

The harvested mature rice grains were dehulled with a rice-huller (Yao jiang JLGJ 2.5, China) and brown rice was milled into fine rice with a rice mill (Osaka MB-RC52 Japan). Finally, they were ground into powder by a cyclone grinder (CT410, FOSS Scino (Suzhou) Co., Ltd., Suzhou, China) and passed through an 80-mesh standard examination sieve to produce rice powder. On a dry basis, the chemical composition of rice flour consists of carbohydrates (about 80 g/100 g, starch (50–60 g/100 g), moisture (about 10–15 g/100 g) proteins (about 7–10 g/100 g), ash (about 3 g/100 g), fats (about 1.5 g/100 g), and pigments [40].

### 3.3. Determination of Rice Amylose and Protein Content

Protein contents (46–11.02) were measured in triplicate according to AACC-approved methods. The amylose content (AACC Method 76–31.01) was determined three times by commercial assay kit (K-AMYL, Megazyme International Ltd., Wicklow, Ireland).

### 3.4. Determination of Eating Quality, Length and Head Rice Rate of Rice Grain

A rice-grain taste analyzer (SATAKE STA1B, Hiroshima, Japan) was used to evaluate the taste value of cooked rice and every sample was determined three times. Freshness score was determined in triplicate by the fresh tester (SATAKE RFDM1B, Hiroshima, Japan). Ten rice grains were randomly picked and measured with a right-angle measurement (Shinwai Sokutei, 12416, Niigata, Japan), and the average was subsequently calculated. Head rice rate was determined in triplicate according to the Chinese National Standard method (GB/T 21719-2008). Pictures of brown rice and milled rice from four varieties were taken by a stereomicroscope (Nikon SMZ800N, Tokyo, Japan); Three parallel runs were performed for each sample.

### 3.5. Determination of Rice Hardness and Stickiness

The hardness and stickiness of N84 and YY-IJHR varieties were measured by a hardness and viscosity tester (SATAKE RHS1A, Hiroshima, Japan). The values of hardness (kgf) and stickness (kgf) were derived from the instrument software (SATAKE RHS1A, Hiroshima, Japan). The hardness and stickiness unit was kgf, where 1 kgf unit is equal to 9.8 Newtons (N).

### 3.6. Determination of Rice Characteristic Aroma Components

Rice samples (5 g, dry basis) from each cultivar were collected in 20-mL headspace bottles and incubated at 80 °C for 15 min before headspace injection, with an injection volume of 500 μL. The injection needle temperature was 85 °C and incubation speed was 500 R/min. The carrier gas was high-purity nitrogen. With flavourspec^®^, the flavor analyzer developed GC-IMS analysis, and each sample was assayed 3 times in parallel. The test conditions of GC-IMS were in accordance with the method of Zhao and Shen [41] and each sample was detected three times. Qualitative analysis of volatile organic components was performed based on the built-in database of GC × IMS Library Search. Quantitative analysis of the corresponding peak area of volatile organic components was normalized with auto-scaling in GC-IMS data.

The contribution of each compound to the fresh rice flavor was evaluated by calculating the relative odor activity value (ROAV). The present study adopted the ROAV method, where the ROAV compound greater than 1 indicated that compounds made a significant contribution to the flavor, and a value of <1 meant a small contribution to the flavor. The component with the greatest contribution to the odor of all samples was defined as ROAV_max_ and given a value of 100, with the other volatile components being calculated according to Equation (1):ROAV_i_ = 100 × C_i_/C_max_) × (T_max_/T_i_).(1)
where C_i_ represented relative concentration (the percentage of each compound peak intensity to total peak intensity of all compounds), and T_i_ indicated the odor threshold of the target volatile components. C_max_ and T_max_ referred to compounds with the maximum odor activity value [26].

### 3.7. Determination of Rice Pasting Properties

The pasting properties of rice flour were determined by a Rapid Visco Analyzer (RVA-TecMaster, Perten Instruments, Warriewood, NSW, Australia). The rice flour samples (3.00 g, based on 14% moisture) were stirred with distilled water (25 mL) in an aluminum vessel. The test parameters were set according to the descriptions of Geng, et al. [42]. The characteristic parameters such as peak viscosity, final viscosity and pasting temperature were obtained by TCW analysis. The unit of characteristic parameters was centipoise (cp), which indicates viscosity. Each sample experiment was measured three times to obtain the end results.

### 3.8. Determination of Rice Cooking Quality

#### 3.8.1. Rice Cooking Expansion and Elongation Rate Measurement

Rice kernels (10 g, dry basis) were accurately weighed and cooked in a small aluminum vessel with 20 mL of water, from which 5 g of cooked rice was loaded into a 10-mL measuring cylinder to determine the volume. The volume of raw rice V0 was measured according to the same method. The cooking expansion rate (%) was calculated according to Equation (2):Cooking expansion rate (%) = 100% × (V_3_ − V_0_)/V_0_.(2)
where W was the total weight of cooked rice; V_1_ was the volume measured immediately after injecting 5 mL of water, V_2_ (V_2_ = V_1_ − 5) was the rice volume; V_3_ (V_3_ = V_2_ × (W/5)) was the volume of total rice. The volume of the raw rice V_0_ was measured according to the same method. 

Actual elongation was measured by dividing the average length of 10 cooked kernels from that of 10 raw rice kernels. Intact rice grains (10 grains) were randomly taken from each sample, and the average length (L_0_) was determined. Ten rice grains were placed into a 20-mL test tube, and allowed to soak for 30 min after the addition of 10 mL distilled water. These soaked rice were subsequently boiled in boiling water for 10 min and the rice grains were removed and placed on filter paper for 60 min to determine the average length (L_1_). Cooking elongation rate was measured according to Equation (3)
Cooking elongation rate (%) = L_1_/L_0_.(3)
where L_1_ represented the average length of cooked rice kernels and L_0_ represented the average length of raw rice kernels [16].

#### 3.8.2. Rice Water Absorption Rate Measurement

Rice grains (10 g, dry basis) were accurately weighed, and the samples were cooked for 30 min in a small aluminum vessel with 20 mL of water in a water bath. The cooked rice kernels were collected and placed on filter paper to drain the surface water. The heating water absorption rate (%) was calculated according to Equation (4)
Heating water absorption rate (%) = 100% × (W_1_ − W_0_)/W_0_.(4)
where, W_0_ represents the weight of 10 g unheated rice kernel and W_1_ represents the weight of cooked rice [16].

To measure the soaked water absorption rate, 3 g of rice were accurately weighed and the samples were soaked for 60 min in a small aluminum vessel with 20 mL of water in a water bath and the aluminum vessel was kept in a constant-temperature water bath of 25 °C. The soaked rice kernels were collected and placed on filter paper to drain the surface water. The soaked water absorption rate (%) was calculated according to Equation (5)
Soaked water absorption rate (%) = 100% × (W_1_ − W_0_)/W_0_.(5)
where W_0_ represented the weight of 3-g unsoaked rice kernels and W_1_ represented the weight of soaked rice.

### 3.9. Determination of Rice FFA and MDA Content

The FFA content of the samples was measured in triplicate according to the Chinese National Standard method (GB/T 29405-2012). In brief, 5 g of rice was added to a conical flask and shaken with 30 mL of ethanol for 1h, the extract was centrifuged, and the supernatant was added to a solution of 1% phenolphthalein ethanol (95%) and titrated with potassium hydroxide ethanol solution to the endpoint. The FFA content was calculated according to the volume of potassium hydroxide ethanol that was consumed by rice grains. The MDA content was determined three times by the kit (A003-1-2, Nanjing Jian Cheng Research Ltd., Nanjing, China).

### 3.10. Determination of Rice Catalase (CAT) and Peroxidase (POD) Activity

The CAT activity was determined in triplicate by a commercial assay kit (A007-1-1, Nanjing Jian Cheng Research Ltd., Nanjing, China). Every milligram of tissue protein that breaks down 1 μmol H_2_O_2_ every second was defined as a unit of catalase activity (U/mg protein). The POD activity was measured in triplicate by commercial assay kit (A084-3-1, Nanjing Jian Cheng Research Ltd., Nanjing, China), and this method used the change in absorbance at 420 nm using the principle that POD catalyzes the reaction of hydrogen peroxide. The amount of enzyme catalyzing 1 mg of substrate per minute per milligram of tissue protein was defined as one unit of enzyme activity.

### 3.11. Statistical Analysis

Data processing was conducted with Microsoft Excel software (Microsoft, Washington, DC, USA) and reported as mean ± standard deviation (SD). One-way ANOVA with Ducan’s tests were performed by IBM SPSS statistical software (version 26.0) at 0.05 significance level. The principal component analysis (PCA) was performed with Origin software (version Pro 2021).

## 4. Conclusions

As an important global starch-based food, rice contributes to nearly 40% of the world’s total caloric intake. YY-IJHR, as a new variety with a high yield and wide planting area, along with its rice quality, starch functionality and storage characteristics, have been attracting increasing attention. The previous study mainly focused on cultivation techniques and yield. However, the rice quality and starch’s functional properties, as well as the storability of YY-IJHR, are important for consumers and industrial applications. Here, three typical YY-IJHR varieties, which included YY12 (biased japonica type YY-IJHR), YY1540 (intermedius type YY-IJHR) and YY15 (biased indica type YY-IJHR), were studied and comparatively analyzed. They were compared with N84 (high-quality conventional japonica rice) in terms of comprehensive rice quality and the functional properties of starch, as well as storability.

The results of this study suggested that the three YY-IJHR varieties all had a greater cooking and eating quality than N84, as they had lower protein and amylose contents. Rice aroma components were revealed by GC-IMS. The alcohol contents in the volatile flavor components of YY-IJHR were generally lower, whereas the content of some aldehydes and esters were higher compared with N84. The volatile aroma components of YY15 were mainly aldehydes and esters, whereas YY12 were mainly aldehydes and ketones. The data from the current study provided the knowledge that YY-IJHR starch has a higher pasting viscosity and digestibility, which may be helpful to common consumers. In addition, both YY-IJHR cultivars’ FFA and MDA contents were lower, which demonstrated that YY-IJHR had higher palatability and storability than N84 in fresh rice and rice stored for 12 months. 

Therefore, YY-IJHR not only has a high yield, but also exhibited a superior grain quality and storability compared with the N84. PCA indicated that grain quality and storability of YY12 and YY15 were similar and performed better than YY1540, while the aroma components and starch functional properties of the YY-IJHR cultivars all showed significant differences. This study will provide valuable information for common consumers when selecting suitable rice varieties and necessary knowledge for physiologists to maximize the yield potential and optimize the application of rice.

## Figures and Tables

**Figure 1 molecules-27-04009-f001:**
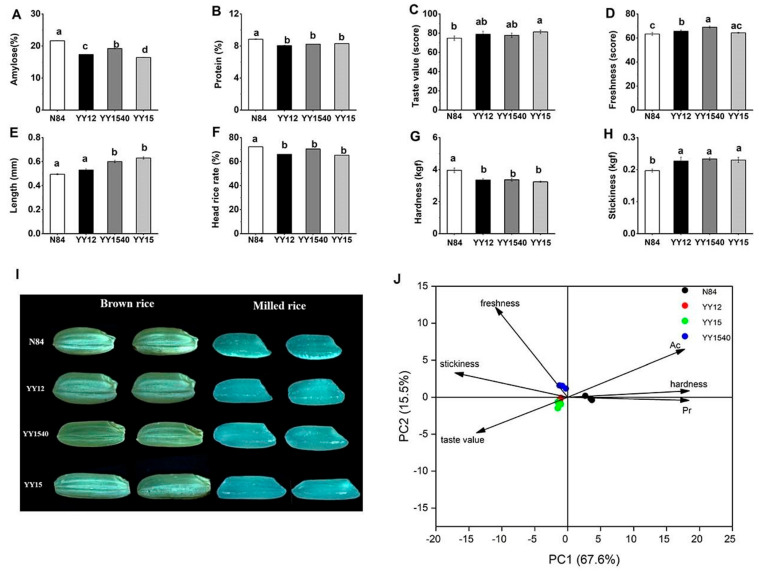
Rice comprehensive quality of YY-IJHR. The amylose content (**A**), protein content (**B**), taste value (**C**), freshness (**D**), length (**E**), head rice rate (**F**), hardness (**G**), and stickiness (**H**), and stereomicrograph (**I**), as well as principal component analysis (PCA) of grain quality (**J**) of N84 and YY-IJHR cultivars. Different lowercase letters indicate significant differences at *p* < 0.05 (Ducan’s test).

**Figure 2 molecules-27-04009-f002:**
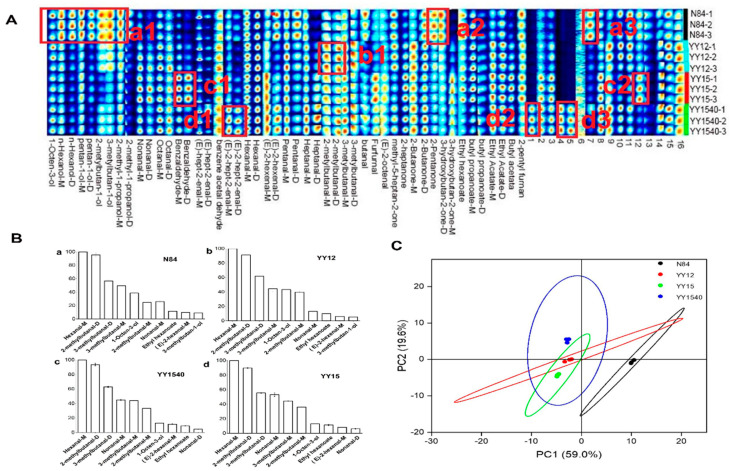
Characteristic of YY-IJHR aroma components. (**A**): Gallery plot fingerprint of volatile components in N84 and YY-IJHR varieties, the characteristics of volatiles among the different rice samples with circled volatile compounds (a1, a2, a3, b1, c1, c2, d1, d2, d3) are presented; (**B**): Fingerprints of the different volatile components and bar graph of the main volatile compounds (ROAV > 1) with the top 10 in four rice varieties; (**C**): PCA of flavor components of N84 and YY-IJHR.

**Figure 3 molecules-27-04009-f003:**
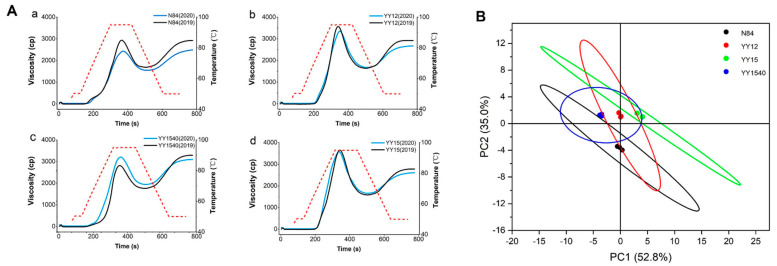
The pasting properties of YY-IJHR. (**A**) Comparison of RVA curves in new rice and rice stored for one year from N84 (**a**) and YY12 (**b**), YY1540 (**c**), YY15 (**d**) as well as PCA of pasting properties (**B**) of N84 and YY-IJHR cultivars.

**Figure 4 molecules-27-04009-f004:**
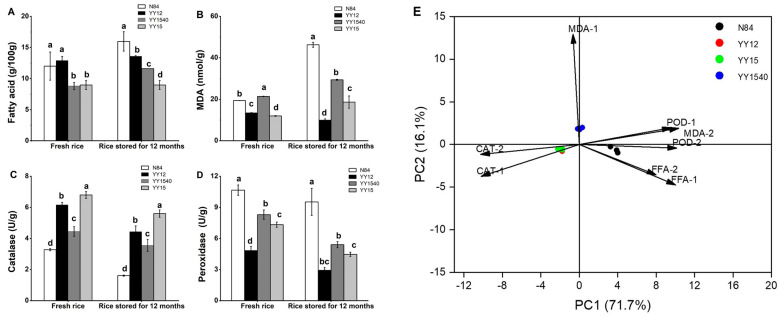
The storability of YY-IJHR. The FFA content (**A**), MDA content (**B**), catalase activity (**C**) and peroxidase activity (**D**) characteristics of YY-IJHR cultivars and N84.In the biplot of PCA of storability (**E**). FFA-1 represented the free fatty acid content of fresh rice, FFA-2 represented the free fatty acid content of rice stored for 12 months; MDA-1 represented the MDA content of fresh rice, MDA-2 represented the MDA content of rice stored for 12 months; CAT-1 represented the CAT activity of fresh rice, CAT-2 represented the CAT activity of rice stored for 12 months; POD-1 represented the POD activity of fresh rice, POD-2 represented the POD activity of rice stored for 12 months. Different lowercase letters indicate significant differences at *p* < 0.05 (Ducan’s test).

**Table 1 molecules-27-04009-t001:** Pasting properties of fresh rice and rice stored for 12 months in N84 and YY-IJHR.

Rice Varieties	N84	YY12	YY1540	YY15
**Fresh Rice**				
PV (cp)	2428 ± 39.3 d	3356 ± 37.6 b	3238 ± 26.1 c	3574 ± 63.9 a
TV (cp)	1549 ± 35.6 b	1677 ± 43.5 b	2040 ± 58.5 a	1660 ± 86.9 b
BD (cp)	878 ± 10.8 d	1678 ± 14.5 b	1199 ± 73.8 c	1915 ± 52.8 a
FV (cp)	2484 ± 35.1 c	2670 ± 41.1 b	3287 ± 68.6 a	2608 ± 73.3 bc
SB (cp)	934 ± 4.1 c	993 ± 2.6 b	1247 ± 13.1 a	948 ± 14.2 c
PT (°C)	75.5 ± 0.1 d	80.3 ± 0.1 b	74.3 ± 0.1 c	81.2 ± 0.2 a
**Rice Stored for 12 Months**				
PV (cp)	2930 ± 42.0 b	3589 ± 52.9 a	2819 ± 72.5 b	3655 ± 9.7 a
TV (cp)	1690 ± 76.0 ab	1640 ± 10.6 ab	1763 ± 61.5 a	1581 ± 38.7 b
BD (cp)	1240 ± 40.9 c	1948 ± 56.0 b	1056 ± 31.7 d	2074 ± 40.8 a
FV (cp)	2926 ± 55.5 c	2918 ± 6.9 b	3286 ± 68.3 a	2784 ± 40.4 b
SB (cp)	1236 ± 20.5 c	1278 ± 16.7 b	1523 ± 8.2 a	1202 ± 8.7 c
PT (°C)	71.1 ± 0.1 c	80.7 ± 0.1 b	86.1 ± 0.4 a	80.2 ± 0.8 b

PV, peak viscosity; FV, final viscosity; TV, trough viscosity; BD, breakdown viscosity; SB, setback viscosity; PT, pasting temperature. cp, centipoises. Means values ± standard deviation; the error bars represent the SD, and different lowercase letters indicate significant differences at *p* < 0.05 (Ducan’s test).

**Table 2 molecules-27-04009-t002:** Fresh rice and rice stored for 12 months’ cooking characteristics for N84 and YY-IJHR.

Rice Varieties	N84	YY12	YY1540	YY15
**Fresh Rice**				
Cooking expansion rate (%)	268 ± 5.7 c	355 ± 1.4 a	364 ± 5.7 a	322 ± 5.7 b
Cooking elongation rate (%)	62.4 ± 0.5 b	77.5 ± 1.4 a	80.2 ± 2.8 a	55.5 ± 7.8 b
Heating water absorption rate (%)	141.6 ± 0.4 b	145.9 ± 1.2 b	143.4 ± 2.8 b	151.8 ± 2.3 a
Soaked water absorption rate (%)	16.1 ± 0.1 c	23.9 ± 0.4 b	26 ± 4.4 ab	30.9 ± 0.4 a
**Rice Stored for 12 Months**				
Cooking expansion rate (%)	460 ± 0.4 a	338 ± 2.8 b	361 ± 1.4 b	367 ± 1.4 b
Cooking elongation rate (%)	53.1 ± 0.1 b	90 ± 2.8 a	67.5 ± 10.1 b	56.6 ± 3.5 b
Heating water absorption rate (%)	344.3 ± 0.2 a	146.4 ± 2.6 b	146.6 ± 0.6 b	148.3 ± 1.8 b
Soaked water absorption rate (%)	14.7 ± 0.1 b	26.0 ± 1.3 a	27.3 ± 4.1 a	30.2 ± 2.8 a

Means values ± standard deviation, *n* = 3. Different lowercase letters indicate significant differences at *p* < 0.05 (Ducan’s test).

## Data Availability

Not Applicable.

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
