# Peer review of "Understanding the Palatability, Flavor, Starch Functional Properties and Storability of Indica-Japonica Hybrid Rice"

_molecules, 2022, doi:10.3390/molecules27134009_

Round 1

Reviewer 1 Report

This work present the functional properties of hybrid rice starch with storing effects.  This manuscript need changes which are mention below.

Abstract: Extensive revision is required in abstract, as the present sentences sounds noisy during reading. There are grammatical errors and need to minimize the sentences in length. Also please delete the first sentence from the abstract. Overall the abstract is not informative enough and need to show the actual picture of the work.

1: Why the functional properties were checked after cooking? Why not at dry form?

Keywords: Good enough

Introduction: Introduction is supported with nice arguments, some information need to address here

1.     Describe in manuscript why this work is different from other rice starch study?

2.     I don’t get any information why this indica-japonica hybrid rice were selected?

3.     No information has been provided for common reader. What was the purpose of such work?

4.     Authors should support their work with arguments why such work is important for rice industries?

5.     Please add/indicate and compare your work with pervious published paper

Methods and materials:

1.     The authors did not mentioned the composition of total rice flour. Define the composition first

2.     How much of protein and other molecules were there in starch? Mention please

3.     I suggests to take control of the popular rice breed, why same the same breed was compared with itself?

4.     Need to define the pasting parameters in details.

5.     How the hardness and stickiness were calculated? No information has been provided

6.     How to control the rice elongation during cooking?

7.     Authors should mention fist elongation or stickiness, which one was important for this study?

Results and discussion: Discussion upon figures and table are fair and according to findings. Add more of information for pasting properties and particularly functional properties of rice starch. 

There is no sense to Figure 3 B. did not explain anything in first look. How to compare figure 3B to figure 3A? Clear this situation in discussion part. Same for figure 2A, B and C.

No difference has been observed for figure 2A or highlight the changes with notation with clear discussion.

I suggest to discuss the figures 3 and 2 with more details. The present arguments are not enough to understand the in-depth insight.

Overall the discussion upon figures 3 and 2 need serious changes.

Conclusion: Change the conclusion as per changes in discussion, add more of application for this research for common reader.

Reviewer 2 Report

Please see the attached comments. 

Round 2

Reviewer 1 Report

Please check for grammar and spellings.